# Mortality rate-dependent variations in antenatal corticosteroid-associated outcomes in very low birth weight infants with 23-34 weeks of gestation: A nationwide cohort study

Jin Kyu Kim[1,2], Jong Hee Hwang[3], Myung Hee Lee[4], Yun Sil Chang[5], Won Soon Park[5]*

1 Department of Pediatrics, Jeonbuk National University School of Medicine, Jeonju, Korea, 2 Biomedical Research Institute of Jeonbuk National University Hospital, Research Institute of Clinical Medicine of Jeonbuk National University, Jeonju, Korea, 3 Department of Pediatrics, Ilsan Paik Hospital, InJe University College of Medicine, Goyang, Korea, 4 Statistic and Data Center, Samsung Medical Center, Seoul, Korea, 5 Department of Pediatrics, Samsung Medical Center, Sungkyunkwan University School of Medicine, Seoul, Korea

☯ These authors contributed equally to this work.
* wonspark@skku.edu

**Data Availability Statement:** Data availability was subjected to the Act on Bioethics and Safety [Law No. 1518, article 18 (Provision of Personal

## Abstract

Antenatal corticosteroid (ACS) administration has been known as one of the most effective treatment in perinatal medicine, but the beneficial effects of ACS may vary not only gestational age, but also the quality of perinatal and neonatal care of the institution. This nationwide cohort study of the Korean Neonatal Network (KNN) data was consisted of <1,500g infants born at 23–34 weeks at 67 KNN hospitals between 2013 and 2017. The 9,142 eligible infants were assigned into two groups–group 1 and 2 <50% and ≥50% mortality rate, respectively, for 23–24 weeks' gestation–reflecting the quality of perinatal and neonatal care. Each group of infants were further stratified into 23–24, 25–26, 27–28, and 29–34 weeks of gestation age. Despite comparable ACS usage between group 1 (82%) and group 2 (81%), the benefits of ACS were only observed in group 1. In the multivariable analyses, infants of group 1 showed significant decrease in mortality and IVH at gestational age 23–24 weeks with ACS use, and the decrease was also seen in early-onset sepsis and respiratory distress syndrome at gestational age of 29–34 weeks while there were no significant decrease in group 2. In this study the overall data was congruent with the previous findings stating that ACS use decreases mortality and morbidity. These results indicate that the improved mortality of infants at 23–24 weeks' gestation reflects the quality improvement of perinatal and neonatal intensive care, which is a prerequisite to the benefits of ACS.

## Introduction

Administration of antenatal corticosteroids (ACS) is one of the oldest and most effective therapies in perinatal medicine. ACS significantly reduce the risk of respiratory distress syndrome

Information)]. Contact for sharing the data or accessing the data can be possible only through the data committee of Korean neonatal network (http://knn.or.kr) and after permitted by the CDC of Korea. Detail contact information was as follows: data access committee: Yun Sil Chang (cys.chang@samsung.com) and ethics committee: Jang Hoon Lee (neopedlee@gmail.com).

**Funding:** This research was supported by a fund (2019-ER7103-00#) from the Research of Korea Centers for Disease Control and Prevention. The funders had no role in study design, data collection, and analysis, decision to publish, or preparation of the manuscript.

**Competing interests:** The authors have declared that no competing interests exist.

(RDS) by fetal maturation, intraventricular hemorrhage (IVH), necrotizing enterocolitis (NEC), early neonatal sepsis, and neonatal death in preterm infants [1,2]. Current guidelines regarding ACS use for risk of preterm birth at 24–34 weeks' gestation is based on limited evidences from dated randomized controlled trials (RCTs) with small sample sizes before 30 weeks' gestation [3], and recent advances in perinatal and neonatal intensive care medicine have resulted in markedly reduced mortality of peri-viable infants of ≤24 weeks' gestation [4–7] and morbidities of more mature extremely preterm infants [8–11]. Nonetheless, as the administration of ACS has nowadays become the standard care for preterm birth, conducting new RCTs for infants ≤34 weeks' gestation might be viewed as unethical. Recently, analyzing very large data sets from a population-based cohort of neonatal networks to confirm the risk/benefit of ACS has thus become best alternative to RCTs [1].

The Korean Neonatal Network (KNN) is a nationwide, multicenter, prospective, web-based cohort registry system for very low birth weight infants (VLBWIs) with a birth weight less than 1,500 g [12,13]. Some of the beneficial effects of ACS have been shown to differ by gestational age [1,2]. In our previous studies, we observed that the mortality rate of the peri-viable infants at 23–24 weeks' gestation reflected the quality of perinatal and neonatal intensive care, and improved the mortality of the infants of 23–24 weeks' gestation was associated with less morbidities in more extremely preterm infants [8,9,14–16]. Moreover, in our previous studies using the KNN data, we observed that the benefits of ACS use was dependent upon not only gestational age, but also the mortality rate of infants at 23–24 weeks' gestation [8,17]. With the above in mind, in this study, we analyzed the nationwide population based KNN cohort data of VLBWIs at 23–34 weeks' gestation to see if there were mortality rate dependent variations in the effect of ACS on neonatal outcomes according to different gestational ages.

## Methods

### Patients

The KNN registry was approved by the institutional review board (IRB) at each participating hospital. Informed written consent to use the data from patients's medical records used for research purpose was obtained from the parents at enrollment by the NICUs participating in the KNN. Informed consent was waived by IRB for infants who died in the delivery room or at the early stage in the NICU before informed consent was able to be obtained for chart review. All methods were carried out in accordance with the IRB-approved protocol and in compliance with relevant guidelines and regulations. The current study utilized KNN database, and each patient's identification code was anonymized to protect the individual's privacy. This study was approved by the institutional Review Board of Jeonbuk National University Hospital.

The database registry of the KNN prospectively registered the clinical information of VLBWIs admitted to 67 voluntarily participating neonatal intensive care units (NICUs) covering >80% of VLBWIs in Korea [12,13]. The enrolment criteria of KNN is registering only VLBWIs actively resuscitated in the delivery room, and admitted to the NICU in this study. Resuscitating infants >24 weeks' gestation is mandatory by law in Korea, but most Korean tertiary NICUs are currently willing to resuscitate infants up to 23 weeks' gestation. Trained staff used a standardized operating procedure to collect demographic and clinical information. Out of 10,399 VLBWIs born between January 1 2013 and December 31 2017 and registered in the database registry of KNN, we collected data on 9,142 infants born at 23 weeks 0 days to 34 weeks 6 days of gestation (Fig 1). We excluded 1,257 infants including 405 infants with gestational age <23 weeks or >34 weeks, 364 ungrouped infants of 25–34 weeks' gestation due to

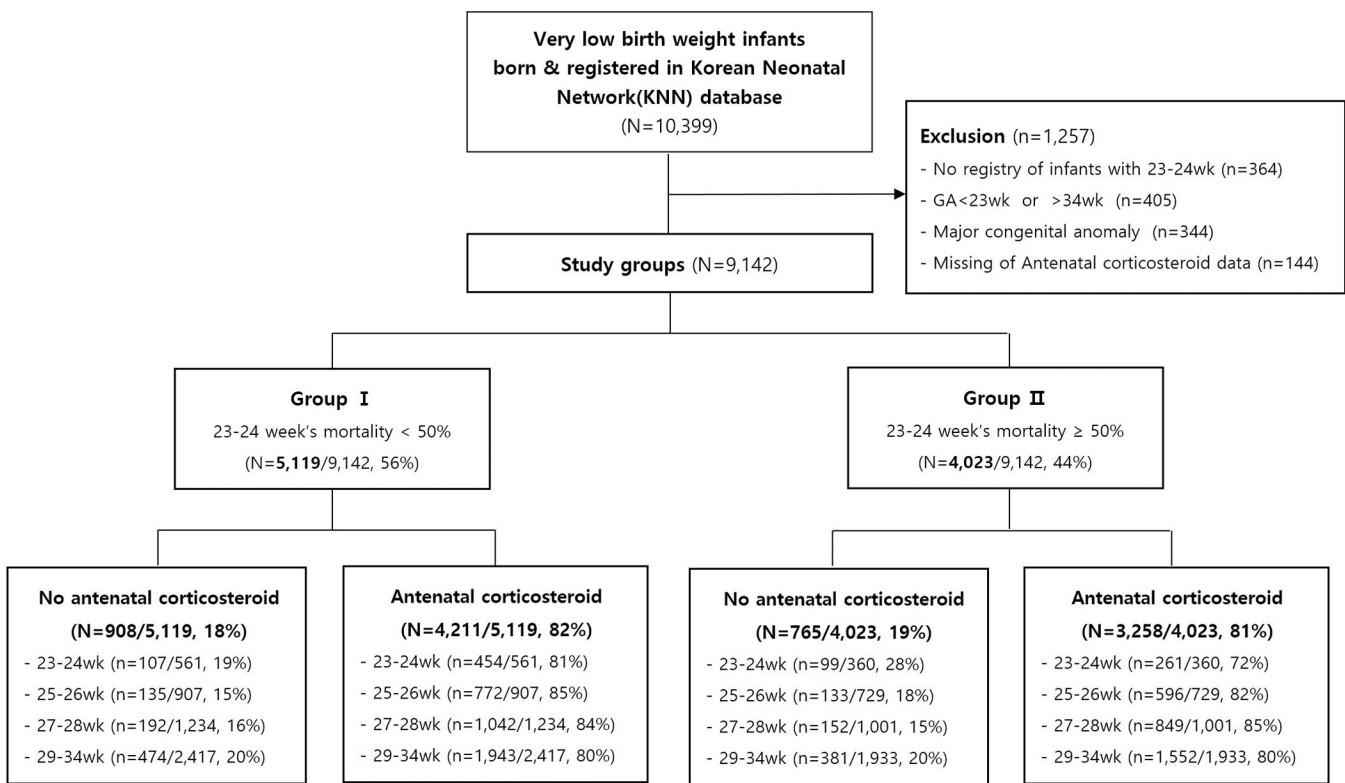

**Fig 1. The wide variability of the mortality rate among infant born at 23–24 weeks gestation from the Korean Neonatal Network included in this study.**

no registry of infants of 23–24 weeks' gestation (9 NICUs), 344 infants with major congenital anomalies, and 144 infants with missing ACS data to reduce the skew of study outcomes because of other causes. We divided the units into two groups according to (1) the baseline mortality of the infants at 23–24 weeks' gestation in this study and (2) the baseline mortality of 50% of the infants at 23–24 weeks' gestation based on previous studies, which show that decreased mortality from 51% to 47% of infants at 23–24 weeks' gestation is associated with increased survival without major morbidity of more mature infants at 25–28 weeks' gestation [18]. Given the wide institutional variation in the mortality rate of these infants, we divided all 9,142 infants with 23–34weeks' gestation into two groups: group 1 included patients from NICUs with a <50% mortality rate for 23–24 weeks' gestation (5,119 patients from 25 NICUs) and group 2 included patients from NICUs with a ≥50% mortality rate for 23–24 weeks' gestation (4,023 patients from 33 NICUs) (Fig 1).

We compared maternal and neonatal variables including gestational age (GA), birth weight, gender, small for gestational age (SGA), mode of delivery, Apgar score at 1 and 5 min, maternal gestational diabetes mellitus (GDM), pregnancy-induced hypertension (PIH), invasive ventilation, noninvasive ventilation, and length of stay between group 1 and 2 in the 23–24, 25–26, 27–28, and 29–34 weeks' gestation subgroups according to ACS use. We compared mortality rates and various major morbidities, including bronchopulmonary dysplasia (BPD), patent ductus arteriosus (PDA), intraventricular hemorrhage (IVH), periventricular leukomalacia (PVL), necrotizing enterocolitis (NEC), retinopathy of prematurity (ROP), and neonatal sepsis between group 1 and 2 in the 23–24, 25–26, 27–28, and 29–34 weeks' gestation subgroups according to ACS use.

## Definitions

We compiled a KNN database operation manual to define patient characteristics. In the manual, GA was determined from the obstetric history based on the last menstrual period. ACS treatment was defined as the administration of any corticosteroid to the mother at any time before delivery to accelerate fetal lung maturity. Chorioamnionitis was confirmed by placental pathology [19], and PROM was defined as the rupture of membranes over 24 hours before the onset of labor. BPD was defined as the use of more than supplemental oxygen at 36 weeks' gestational age, corresponding to moderate to severe BPD using the severity-based definition for BPD of the National Institutes of Health consensus [20]. Symptomatic PDA was defined as clinical symptoms of PDA, such as ventilator dependence, deteriorating respiratory status, increasing recurrent apnea, pulmonary hemorrhage and hypotension. IVH was defined as grade ≥3 according to the classification of Papile et al [21]. PVL was defined as cystic PVL based on either head ultrasound or cranial magnetic resonance imaging scans performed at ≥2 weeks of age. NEC was defined as ≥stage 2b according to the modified Bell criteria [22]. Early sepsis was defined as a positive blood culture less than 7 days from birth in symptomatic infants suggestive of septicemia and more than 5 days of antibiotic treatment [8,17]. ROP was defined as any ROP that needs anti-vascular endothelial growth factor and/or laser ablative and/or surgical treatment to prevent visual loss [23].

## Statistical analysis

The characteristics of the study participants and their prenatal and neonatal morbidities are described as mean ± standard deviation for continuous variables and as numbers and proportions for binary and categorical variables. Continuous variables were compared using the t-test or Wilcoxon rank-sum test. Categorical variables are presented as percentages and frequencies and compared using the chi-square or Fisher's exact test. Logistic regression was used to estimate the odds ratio (OR) with 95% confidence interval (CI) with adjustment for GA, Apgar score at 5 minutes, SGA, cesarean section, multiple pregnancies, inborn, and PIH. A $p$-value $<0.05$ was considered to be statistically significant. Statistical analyses were performed using STATA version 14.0 (STATA Corp., College, TX, USA).

## Results

### ACS use

While there was no significant difference in the overall ACS use between group 1 (82%, 4,211/5,119) and group 2 (81%, 3,258/4,023), the ACS use in 23–24 weeks' gestation subgroups in group 1 (81%, 454/561) was slightly but significantly higher than in group 2 (72%, 261/360) ($p = 0.003$) (Fig 2).

### Demographic and perinatal characteristics

Demographic and perinatal characteristics of the VLBWIs according to ACS use in each group and subgroup are shown in Table 1. The overall GA, birth weight, and SGA were significantly lower, and Apgar scores at 1 & 5 min, cesarean section, multiple pregnancy, inborn, maternal GDM, and chorioamnionitis were significantly higher with ACS use than without. For group comparison, GA, birth weight, Apgar scores at 1 & 5 min, and cesarean section were significantly lower, and SGA, multiple pregnancies, and chorioamnionitis were significantly higher in group 1 with ACS use than in group 2 with ACS use.

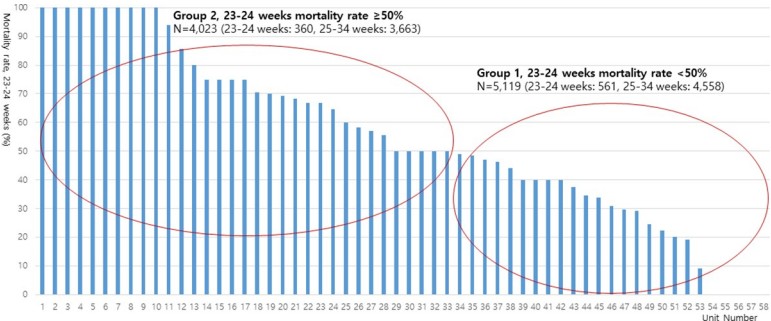

**Fig 2. Flowchart showing the study population.** This study enrolled 10,399 VLBWI. We excluded 1,257 infants, and these infants were categorized into 2 groups on the bases of their mortality (<50% and ≥50%) at 23–24 weeks of gestation.

## Mortality and morbidities

Table 2 demonstrates the mortality and morbidities according to ACS use in each group and GA subgroups. While the overall mortality rates were significantly improved with ACS use in both group 1 and group 2, in subgroup analysis, the mortality rate in group 1 only at 23–24 weeks' gestation was significantly improved with ACS use. The mortality rates of group 2 in all gestational age subgroups were significantly higher than in group 1, regardless of ACS use. While the overall prevalence of IVH (>3) was reduced with ACS use, subgroup analysis revealed that IVH was significantly reduced in group 1 at 23–24 weeks' gestation and the prevalence of periventricular leukomalacia was significantly reduced with ACS use in group 1 at 25–26 weeks' gestation. Otherwise the overall prevalence of NEC was significantly increased with ACS use in group 1 at 23–26 weeks' gestation.

In group 2 the prevalence of symptomatic PDA increased at 25–26 weeks' gestation and the RDS and sepsis at 29–34 weeks' gestation with ACS use increased in group 2.

## Adjusted OR for mortality and morbidities

Fig 3 shows the adjusted odds ratios (OR) and 95% confidence intervals (CI) for the mortality and major morbidities associated with ACS use stratified by GA group. Adjusted variables were gestational age, Apgar score at 5 minute, SGA, cesarean section, multiple pregnancies, inborn, and PIH. In the infants with 23–24 weeks' gestation, the adjusted OR for mortality (0.54, 95% CI; 0.34–0.87) and IVH (0.42, 95% CI; 0.25–0.69) with ACS use was significantly reduced in group 1 and total weeks' gestation but not 2. NEC with ACS use was significantly increased in group 1 infants at 23–24 weeks' gestation (2.44, 95% CI; 1.19–5.02) and total weeks' gestation (1.75, 95% CI 1.07–2.86). In the infants with 25–26 weeks' gestation, the adjusted OR for periventricular leukomalacia (0.57, 95% CI; 0.33–0.96) with ACS use was significantly reduced in group 1 but not 2. Symptomatic PDA (1.54, 95% CI; 1.01–2.34) with ACS use were significantly increased in group 2 but not 1. In the infants with 29–34 weeks' gestation, the adjusted OR for RDS (0.62, 95% CI; 0.48–0.81) and early onset neonatal sepsis (0.30, 95% CI 0.14–0.62) with ACS use was significantly reduced in group 1.

## Discussion

Given that ACS therapy is now a "standard of care" and widely used, large cohort studies rather than the currently infeasible RCTs might be the best to address a number of unanswered questions for its use. Recent cohort studies of very large data sets from neonatal networks and clinical study groups have shown that the beneficial effects of ACS differ by gestational age

**Table 1. Comparison of demographic and perinatal characteristics.**

| Variables | 23–24 weeks (N = 921) | | 25–26 weeks (N = 1,636) | | 27–28 weeks (N = 2,235) | | 29–34 weeks (N = 4,350) | | Total (N = 9,142) | |
|---|---|---|---|---|---|---|---|---|---|---|
| | No ACS | ACS | No ACS | ACS | No ACS | ACS | No ACS | ACS | No ACS | ACS |
| | n = 206 | n = 715 | n = 268 | n = 1,368 | n = 344 | n = 1,891 | n = 855 | n = 3,495 | n = 1,673 | n = 7,469 |
| **Group 1** | 107 (19%) | 454 (81%) | 135 (15%) | 772 (85%) | 192 (16%) | 1,042 (84%) | 474 (20%) | 1,943 (80%) | 908 (18%) | 4,211 (82%) |
| Gestational age (weeks) | $24^{0/7}\pm0^{0/7}$ | $24^{1/7}\pm0^{4/7}$[a] | $26^{0/7}\pm0^{4/7}$ | $26^{0/7}\pm0^{4/7}$ | $28^{0/7}\pm0^{4/7}$ | $28^{0/7}\pm0^{4/7}$ | $31^{3/7}\pm1^{6/7}$ | $30^{6/7}\pm1^{3/7}$[a] | $29^{0/7}\pm3^{1/7}$ | $28^{4/7}\pm2^{5/7}$[a] |
| Birth weight (g) | 648.2±123.0 | 648.5±113.5 | 861.7±167.8 | 819.7±168.2[a] | 1,080.0±197.1 | 1,059.4±208.8 | 1,278.0±189.6 | 1,244.4±207.1[a] | 1,100.0±286.4 | 1,056.5±285.8[a] |
| 1-min Apgar score | 2.4±1.6 | 3.0±1.7[a] | 3.3±1.7 | 3.6±1.8[a] | 3.8±1.9 | 4.5±1.8[a] | 5.3±2.0 | 5.5±1.8[a] | 4.3±2.2 | 4.6±2.0[a] |
| 5-min Apgar score | 4.4±2.0 | 5.3±2.0[a] | 5.9±2.1 | 6.0±1.9 | 6.2±1.8 | 6.7±1.6[a] | 7.3±1.7 | 7.6±1.4[a] | 6.6±2.0 | 6.8±1.8[a] |
| Male sex, n(%) | 55(51) | 225(50) | 77(57) | 409(53) | 98(51) | 566(54) | 232(49) | 958(49) | 462(51) | 2,158(51) |
| Small for gestational age, n(%) | 17(16) | 70(15) | 15(11) | 125(16) | 18(9) | 126(12) | 218(46) | 729(38)[a] | 268(30) | 1,050(25)[a] |
| Cesarean section, n(%) | 47(44) | 293(65)[a] | 92(68) | 586(76) | 139(72) | 803(77) | 404(85) | 1,617(83) | 682(75) | 3,299(78)[a] |
| Multiple pregnancy, n(%) | 29(27) | 167(37) | 36(27) | 255(33) | 52(27) | 376(36)[a] | 176(37) | 810(42) | 293(32) | 1,608(38)[a] |
| Inborn, n(%) | 98(92) | 445(98)[a] | 131(97) | 750(97) | 175(91) | 1,015(97)[a] | 454(96) | 1,919(99)[a] | 858(94) | 4,129(98)[a] |
| Maternal Gestational DM, n(%) | 0(0) | 16(4) | 13(10) | 48(6) | 14(7) | 87(8) | 31(7) | 202(10)[a] | 58(6) | 353(8)[a] |
| PIH, n(%) | 4(4) | 24(5) | 12(9) | 79(10) | 24(13) | 144(14) | 128(27) | 569(29) | 168(19) | 816(19) |
| Chorioamnionitis, n(%) | 64(70) | 259(62) | 52(43) | 366(52) | 69(42) | 413(43) | 102(24) | 424(25) | 287(36) | 1,462(38) |
| Length of stay (day) | 75.5±66.3 | 99.3±64.4[a] | 85.2±51.5 | 94.7±48.7[a] | 72.7±29.9 | 74.3±33.2 | 46.7±21.6 | 50.6±24.5[a] | 61.3±39.8 | 69.8±42.6[a] |
| Invasive ventilator (day) | 34.5±28.9 | 44.6±35.9[a] | 27.8±30.5 | 29.2±31.1 | 13.4±27.5 | 12.3±24.7 | 3.4±9.2 | 3.5±10.4 | 12.8±24.0 | 14.8±26.7 |
| Noninvasive ventilator (day) | 16.4±19.7 | 27.5±27.2[a] | 26.7±32.4 | 30.4±25.6[a] | 21.2±17.1 | 23.6±19.1 | 8.1±12.0 | 10.1±14.6[a] | 14.6±19.8 | 19.0±21.5[a] |
| **Group 2** | 99 (28%) | 261 (72%) | 133 (18%) | 596 (82%) | 152 (15%) | 849 (85%) | 381 (20%) | 1,552 (80%) | 765 (19%) | 3,258 (81%) |
| Gestational age (weeks) | $23^{6/7}\pm0^{4/7}$ | $24^{1/7}\pm0^{4/7}$[a] | $25^{6/7}\pm0^{4/7}$ | $26^{0/7}\pm0^{4/7}$ | $28^{0/7}\pm0^{4/7}$ | $28^{0/7}\pm0^{4/7}$ | $31^{5/7}\pm1^{5/7}$ | $30^{6/7}\pm1^{3/7}$[a] | $28^{6/7}\pm3^{2/7}$ | $28^{5/7}\pm2^{4/7}$[b] |
| Birth weight (g) | 644.8±121.0 | 656.3±107.7 | 875.4±140.6 | 835.0±151.2[a] | 1,104.0±192.7 | 1,082.5±195.3[b] | 1,299.7±168.2 | 1,267.0±189.2[a,b] | 1,102.3±286.8 | 1,091.0±272.0[b] |
| 1-min Apgar score | 2.4±1.5 | 3.0±1.7[a] | 3.1±2.0 | 3.9±1.7[a,b] | 4.0±1.9 | 4.7±1.8[a,b] | 5.5±2.1 | 5.6±1.8 | 4.4±2.3 | 4.9±1.9[a,b] |
| 5-min Apgar score | 4.4±2.0 | 5.3±2.0[a] | 5.4±2.1[b] | 6.4±1.6[a,b] | 6.3±1.8 | 6.9±1.6[a,b] | 7.5±1.6 | 7.6±1.4 | 6.5±2.1 | 7.0±1.7[a,b] |
| Male sex, n(%) | 41(41) | 141(54)[a] | 65(49) | 319(54) | 86(57) | 452(53) | 182(48) | 736(47) | 374(49) | 1,648(51) |
| Small for gestational age, n(%) | 14(14) | 34(13) | 6(5)[b] | 69(12)[a] | 14(9) | 78(9)[b] | 175(46) | 526(34)[a,b] | 209(27) | 707(22)[a,b] |
| Cesarean section, n(%) | 45(45) | 176(67)[aa] | 80(60) | 445(75)[a] | 106(70) | 668(79)[a] | 300(79)[b] | 1,325(85)[a] | 531(69)[b] | 2,614(80)[a,b] |
| Multiple pregnancy, n(%) | 34(34) | 86(33) | 35(26) | 168(28) | 45(30) | 241(28)[b] | 132(35) | 569(37)[b] | 246(32) | 1,064(33)[b] |
| Inborn, n(%) | 93(94) | 259(99)[a] | 122(92) | 592(99)[a,b] | 129(85) | 839(99)[a,b] | 363(95) | 1,539(99)[a] | 707(92) | 3,229(99)[a,b] |
| Maternal Gestational DM, n(%) | 2(2) | 8(3) | 8(6) | 37(6) | 9(6) | 82(10) | 38(10) | 146(9) | 57(7) | 273(8) |
| PIH, n(%) | 5(5) | 16(6) | 6(5) | 49(8) | 16(11) | 115(14) | 109(29) | 450(29) | 136(18) | 630(19) |
| Chorioamnionitis, n(%) | 26(41)[b] | 101(53)[b] | 49(46) | 215(46) | 26(23)[b] | 239(37)[a,b] | 57(20) | 279(23) | 158(28)[b] | 834(33)[a,b] |
| Length of stay (day) | 42.6±58.5[b] | 54.8±68.0[a,b] | 79.3±56.8 | 80.7±57.1[b] | 73.4±39.1 | 73.9±33.2 | 49.5±25.1 | 52.5±21.2[a,b] | 58.5±42.5 | 63.4±40.2[a,b] |
| Invasive ventilator (day) | 22.7±28.0[b] | 32.0±41.1[a,b] | 28.6±33.7 | 32.3±41.4 | 13.6±17.5 | 15.1±23.9[b] | 5.0±18.1 | 4.5±12.3[b] | 13.1±24.6 | 14.6±28.3 |

(Continued)

**Table 1.** (Continued)

| Variables | 23–24 weeks(N = 921) | | 25–26 weeks (N = 1,636) | | 27–28 weeks (N = 2,235) | | 29–34 weeks (N = 4,350) | | Total (N = 9,142) | |
|---|---|---|---|---|---|---|---|---|---|---|
| | No ACS | ACS | No ACS | ACS | No ACS | ACS | No ACS | ACS | No ACS | ACS |
| | n = 206 | n = 715 | n = 268 | n = 1,368 | n = 344 | n = 1,891 | n = 855 | n = 3,495 | n = 1,673 | n = 7,469 |
| Noninvasive ventilator (day) | 12.2 ±26.3[b] | 12.9 ±25.7[b] | 22.7±22.9 | 25.1±26.0[b] | 22.3±19.4 | 25.5±20.2[a,b] | 8.4±11.1 | 12.2±14.0[a,b] | 14.2±18.9 | 18.1±20.4[a] |

ACS, antenatal corticosteroid; PIH, pregnancy induced hypertension.

a. p<0.05 compared with No ACS.

b. p<0.05 compared with Group I.

[1,2,24–26]. Travers et al. reported that in 117,941 infants from 23 to 34 weeks' gestation, while ACS use was associated with lower mortality and morbidity at most gestations, the benefits were greatest in infants at gestations of 23–24 weeks [1]. In a prospective cohort study of 13,406 infants born between 23 and 32 weeks' gestation, ACS was associated with improved survival in infants born between 24 and 29 weeks' gestation [25]. In a retrospective analysis of 11,607 infants born at 22 to 33 weeks' gestation, ACS improved the survival of infants at 22–27 weeks' gestation and decreased RDS and severe IVH at 24–29 weeks' gestation [7]. In 29,932 infants receiving postnatal life support at 22–25 weeks' gestation, ACS was associated with improved survival and survival without major morbidities at 22–25 weeks' gestation [5]. In the present nationwide population-based KNN cohort study of 9,142 VLBWIs from 23 to 34 weeks' gestation, ACS was associated with improved survival and reduced severe IVH at 23–24 weeks' gestation and reduced RDS and early onset sepsis at 29–34 weeks' gestation. Taken together, these findings suggest that while the benefits of ACS use were observed in preterm infants up to 34 weeks' gestation, the lowest gestational age for ACS use could be extended to at least 23 weeks' gestation.

In our previous studies, the improved mortality rate of infants at 23–24 weeks' gestation indicated the quality improvement of perinatal and NICU care, including better delivery room resuscitation [8,9,14–16]. In the present study, the benefits of ACS such as decreased mortality and severe IVH, especially in infants at 23–24 weeks' gestation, and decreased RDS and sepsis at 29–34 weeks' gestation were observed in group 1 with mortality <50% but not in group 2 with mortality ≥50% in infants at 23–24 weeks' gestation. These findings suggest that the beneficial effects of ACS use differ not only by GA but also by quality of perinatal and NICU care.

Concurrent provision of ACS use and resuscitation following extremely preterm birth increased infant survival and survival without morbidities [4,5,7,27]. However, despite our KNN data including only actively resuscitated VLBWIs, and comparable ACS use rates of 82% and 81% between group 1 and 2, mortality rates throughout 23–34 weeks' gestation were significantly higher in group 2 than in group 1, regardless of ACS use in this study. For international comparison of 10 national neonatal networks, the lowest mortality rate of VLBWIs were observed in the Japanese neonatal research network, despite having the lowest ACS (53.7%) use compared with other networks (75–94%) [28,29]. Overall, these results suggest that quality improvement of perinatal and NICU care outweigh the beneficial effects of ACS use for improved mortality and morbidities of VLBWIs [30–33].

In contrast to other studies showing decreased prevalence of NEC [2,34,35], the prevalence of NEC in group 1 infants at 23–24 weeks' gestation compared with group 2 infants at 23–24 weeks' gestation was significantly increased in this study. However, the prevalence of NEC with ACS use in group 2 was significantly lower than that in Group 1, and death or NEC with

**Table 2. Comparison of mortality and morbidities.**

| Morbidities | 23–24 weeks (N = 921) | | 25–26 weeks (N = 1,636) | | 27–28 weeks (N = 2,235) | | 29–34 weeks (N = 4,350) | | Total (N = 9,142) | |
|---|---|---|---|---|---|---|---|---|---|---|
| | No ACS | ACS | No ACS | ACS | No ACS | ACS | No ACS | ACS | No ACS | ACS |
| | n = 206 | n = 715 | n = 268 | n = 1,368 | n = 344 | n = 1,891 | n = 855 | n = 3,495 | n = 1,673 | n = 7,469 |
| **Group 1** | 107 (19%) | 454 (81%) | 135 (15%) | 772 (85%) | 192 (16%) | 1,042 (84%) | 474 (20%) | 1,943 (80%) | 908 (18%) | 4,211 (82%) |
| Mortality, n(%) | **51(48)** | **145(32)**[a] | 27(20) | 119(15) | 14(7) | 62(6) | 12(3) | 31(2) | **104(12)** | **357(8)**[a] |
| Air leak syndrome, n(%) | 16(15) | 69(15) | 14(10) | 81(10) | 11(6) | 38(4) | 12(3) | 28(1) | 53(6) | 216(5) |
| Respiratory distress syndrome, n(%) | 105(98) | 447(98) | 133(99) | 751(97) | 183(95) | 951(91) | 294(62) | 1,163(60) | 715(79) | 3,312(79) |
| Bronchopulmonary dysplasia (≥moderate), n(%) | 44(77) | 242(76) | 62(57) | 353(53) | 60(33) | 319(32) | 56(12) | 245(13) | 222(28) | 1,159(30) |
| Symptomatic patent ductus arteriosus, n (%) | 60(58) | 259(62) | 71(55) | 378(53) | 73(38) | 379(38) | 73(16) | 346(18) | 277(31) | 1,362(34) |
| Intraventricular hemorrhage (≥grade 3), n(%) | **42(49)** | **130(31)**[a] | 23(19) | 101(14) | 15(8) | 79(8) | 9(2) | 44(2) | 89(10) | 354(9) |
| Periventricular leukomalacia, n(%) | 17(20) | 66(16) | **22(18)** | **82(11)**[a] | 17(9) | 85(8) | 19(4) | 103(5) | 75(9) | 336(8) |
| Necrotizing enterocolitis (≥stage 2), n(%) | **12(12)** | **93(21)**[a] | **7(5)** | **87(11)**[a] | 11(6) | 68(7) | 11(2) | 38(2) | **41(5)** | **286(7)**[a] |
| Sepsis (total), n(%) | 40(38) | 192(42) | 41(31) | 240(31) | 39(20) | 227(22) | 44(9) | 171(9) | 164(18) | 830(20) |
| Early sepsis within 7 days of life, n(%) | 11(11) | 47(10) | 7(5) | 41(5) | 9(5) | 52(5) | **13(3)** | **20(1)**[a] | 40(4) | 160(4) |
| Retinopathy of premature (operation), n (%) | 33(56) | 173(53) | 22(20) | 153(23) | 13(7) | 57(6) | 3(1) | 20(1) | 71(9) | 403(11) |
| **Group 2** | 99 (28%) | 261 (72%) | 133 (18%) | 596 (82%) | 152 (15%) | 849 (85%) | 381 (20%) | 1,552 (80%) | 765 (19%) | 3,258 (81%) |
| Mortality, n(%) | 75(76)[b] | 186(71)[b] | 46(35)[b] | 206(35)[b] | 19(13) | 88(10)[b] | 13(3) | 57(4)[b] | **153(20)**[b] | **537(16)**[a,b] |
| Air leak syndrome, n(%) | 19(19) | 46(18) | 17(13) | 53(9) | 6(4) | 36(4) | 8(2) | 30(2) | 50(6) | 165(5) |
| Respiratory distress syndrome, n(%) | 96(97) | 253(97) | 128(96) | 586(98) | 145(95) | 809(95)[b] | **210(55)**[b] | **1,038 (67)**[a,b] | **579(76)** | **2,686 (82)**[a,b] |
| Bronchopulmonary dysplasia (≥moderate), n(%) | 19(76) | 62(76) | 57(63) | 250(62)[b] | 50(38) | 284(37)[b] | 55(15) | 246(16)[b] | 181(29) | 842(31) |
| Symptomatic patent ductus arteriosus, n (%) | 32(36)[b] | 90(38)[b] | **47(36)**[b] | **282(48)**[a] | 55(36) | 302(36) | 56(15) | 267(17) | **190(25)**[b] | **941(29)**[a,b] |
| Intraventricular hemorrhage (≥grade 3), n(%) | 27(40) | 79(35) | 30(25) | 110(19)[b] | 14(10) | 61(7) | 7(2) | 26(2) | 78(11) | 276(9) |
| Periventricular leukomalacia, n(%) | 4(6)[b] | 20(9)[b] | 18(15) | 67(12) | 14(10) | 65(8) | 25(7) | 70(5) | 61(9) | 222(7) |
| Necrotizing enterocolitis (≥stage 2), n(%) | 14(14) | 35(14)[b] | 18(14)[b] | 74(13) | 12(8) | 66(8) | 10(3) | 59(4)[b] | 54(7)[b] | 234(7) |
| Sepsis (total), n(%) | 30(31) | 109(42) | 49(37) | 230(39)[b] | 39(26) | 223(26)[b] | **36(9)** | **237(15)**[a,b] | **154(20)** | **799(25)**[a,b] |
| Early sepsis within 7 days of life, n(%) | 6(6) | 35(14) | 11(8) | 44(7) | 5(3) | 40(5) | 10(3) | 50(3)[b] | 32(4) | 169(5)[b] |
| Retinopathy of premature (operation), n (%) | 12(40) | 40(45) | 25(27) | 97(23) | 6(5) | 46(6) | 4(1) | 15(1) | 47(8) | 198(7)[b] |

ACS, Antenatal corticosteroids.

a. p<0.05 compared with No ACS.

b. p<0.05 compared with Group.

ACS vs. without NEC in group 1 was comparable in this study. In concordance with our data, Travers et al. reported that while infants exposed to ACS had lower rate of death or NEC, rates of NEC was significantly higher in the infants with ACS use at 23 weeks' gestation than without ACS use [1]. In other studies also found that exposure to ACS was associated with an increased risk for NEC in extremely preterm infants [36–38]. Overall, these findings suggest that the increased NEC following administration of ACS observed only in group 1 simply reflect improved survival of extremely preterm infants at the highest risk of developing NEC.

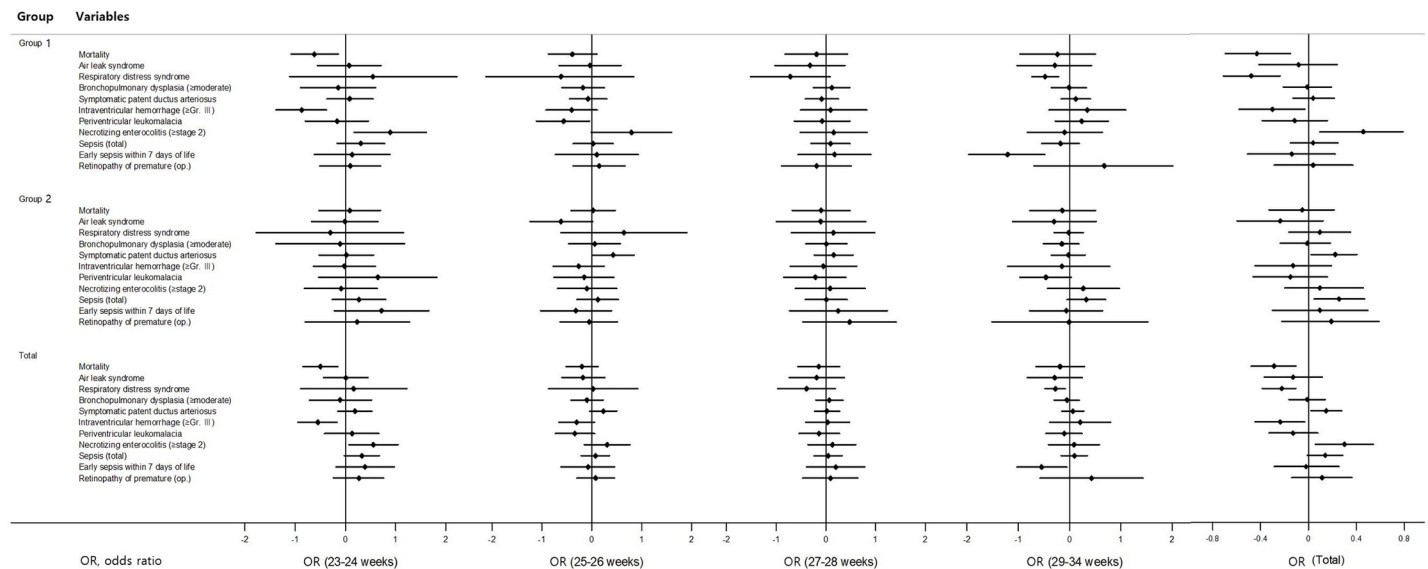

**Fig 3. Adjusted odds ratio of mortality and morbidities associated with antenatal corticosteroid use (95% confidence interval).**

In the present study, the detrimental effects of ACS use such as increased overall prevalence of symptomatic PDA and sepsis observed in group 2, contradictory to beneficial effects such as improved mortality, RDS and severe IVH observed in group 1, are difficult to explain. Stoll et al. reported that improved survival of extremely preterm infants, especially those born at 23–24 weeks' gestation, was accompanied by increased survival without major morbidities in more mature infants at 25–28 weeks' gestation [39]. In our previous studies, improved survival of peri-viable infants at 23–24 weeks' gestation significantly reduced morbidities such as BPD and sepsis and thus increased intact survival in more mature infants at 25–26 weeks' gestation [8,9,14]. Taken together, these results suggest that improved quality of perinatal and NICU care such as well-experienced and skillful neonatologist, better early admission care, and prevention of complication improved mortality of peri-viable infants at 23–24 weeks gestation. Therefore, higher quality of clinical care can be a prerequisite to the beneficial effects associated with ACS use.

The strength of this study was that it included a prospective nationwide population-based study of VLBWIs between 23–34 weeks' gestation with/without ACS use. In addition, as our study enrolled only actively resuscitated VLBWIs, the possibility of restricted or withheld postnatal care in the infants not exposed to ACS is minimal [11,26,40,41]. However, no available data regarding timing, partial or complete, or single/multiple ACS use are limitations of this study.

In conclusion, in a nationwide prospective cohort study of VLBWIs between 23 and 34 weeks' gestation, the benefits of ACS use including improved mortality and morbidities of IVH and RDS were observed only in lower mortality infants at 23–24 weeks' gestation. Furthermore, detrimental effects of ACS use including a higher rate of sepsis and symptomatic PDA were observed in higher mortality infants at 23–24 weeks' gestation. These findings implicate that quality improvement of perinatal and NICU care with the resultant improved survival of peri-viable infants at 23–24 weeks' gestation are prerequisite to the benefits of ACS use.

## Author Contributions

**Conceptualization:** Yun Sil Chang, Won Soon Park.

**Data curation:** Jin Kyu Kim, Jong Hee Hwang, Myung Hee Lee.

**Formal analysis:** Jin Kyu Kim, Jong Hee Hwang.

**Funding acquisition:** Won Soon Park.

**Methodology:** Jin Kyu Kim.

**Project administration:** Myung Hee Lee.

**Software:** Myung Hee Lee.

**Supervision:** Yun Sil Chang, Won Soon Park.

**Writing – original draft:** Jin Kyu Kim, Jong Hee Hwang, Won Soon Park.

**Writing – review & editing:** Jin Kyu Kim, Jong Hee Hwang, Myung Hee Lee, Yun Sil Chang, Won Soon Park.

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
