## [Decision Letter · Decision Letter 0]

27 Jul 2020

PONE-D-20-16654

Mortality rate-dependent variations in antenatal corticosteroid-associated outcomes in very low birth weight infants with 23-34 weeks of gestation: A nationwide cohort study

PLOS ONE

Dear Dr. Won Soon Park

Thank you for submitting your manuscript to PLOS ONE. After careful consideration, we feel that it has merit but does not fully meet PLOS ONE’s publication criteria as it currently stands. Therefore, we invite you to submit a revised version of the manuscript that addresses the points raised during the review process.

We look forward to receiving your revised manuscript.

Kind regards,

Georg M. Schmölzer

Academic Editor

PLOS ONE

Journal Requirements:

2. In your ethics statement in the manuscript and in the online submission form, please provide additional information about the patient records used in your retrospective study. Specifically, please ensure that you have discussed whether all data were fully anonymized before you accessed them and/or whether the IRB or ethics committee waived the requirement for informed consent. If patients provided informed written consent to have data from their medical records used in research, please include this information.

"This research was supported by a fund (2019-ER7103-00#) from the Research of Korea Centers for

Disease Control and Prevention. There are ethical restrictions on sharing a deidentified data set

unless permitted by the CDC of Korea.".

i) We note that you have provided funding information that is not currently declared in your Funding Statement. However, funding information should not appear in the Acknowledgments section or other areas of your manuscript. We will only publish funding information present in the Funding Statement section of the online submission form.

ii) Please remove any funding-related text from the manuscript and let us know how you would like to update your Funding Statement. Currently, your Funding Statement reads as follows:

 "The funders had no role in study design, data collection and analysis, decision to publish, or preparation of the manuscript.".

Reviewers' comments:

Reviewer's Responses to Questions

**Comments to the Author**

1. Is the manuscript technically sound, and do the data support the conclusions?

Reviewer #1: Partly

2. Has the statistical analysis been performed appropriately and rigorously? 

Reviewer #1: Yes

3. Have the authors made all data underlying the findings in their manuscript fully available?

Reviewer #1: Yes

4. Is the manuscript presented in an intelligible fashion and written in standard English?

Reviewer #1: Yes

5. Review Comments to the Author

Reviewer #1: This article by Park et al., on Mortality rate-dependent variations in antenatal corticosteroid-associated outcomes in very low birth weight infants with 23-34 weeks of gestation: A nationwide cohort study is an important topic with regard to use of antenatal steroids and assess why their effect is different at different GA based on mortality.

Overall, the paper is well written and is clear.

Introduction: provides sufficient background information.

Methods: The data is from 67 neonatal units from KNN registry. Rest of the methodology section is clear.

Results:

Authors have indicated ACS use was 81% in group1 for GA 23-24wk compared to group 2 was 72%.

Mortality was significantly higher for 23-24 wk. in-group 2 compared to group 1 and rest of the GA the mortality rates remained high in-group 2.

The incidence of IVH reduced in 23-24 wk., reduced PVL at 25-26 wk., decreased RDS, early onset sepsis at 29-34 wk. and increased NEC at 23-26 wk. in-group 1 with ACS use.

Symptomatic PDA at 25-26 week, RDS and sepsis at 29-34 wk. was higher in-group 2 with ACS use.

BPD/ROP rates were not different between the groups.

They have not given details of ACS with regard to doses; partial or complete between the groups on infants who received ACS

Discussion:

Authors have quoted important studies showing benefits of ACS use on mortality and morbidities in infants born at 22-34 wk. gestation.

With improved IVH, RDS, PVL, early onset sepsis in Group 1 with mortality <50% with ACS use.

Authors make a good point that with ACS use, some of the morbidities are different when compared based on mortality rate for each gestational age category.

Most morbidities were much improved in infants in group 1 and some morbidities were better (NEC) in group 2 with ACS use

They mentioned in the method section that they were unable to group them based on the neonatal units because of wide institutional variation in the mortality rate in these infants(23-24 wk). Their conclusion that the beneficial effects of ACS use differ not only GA but also by quality of perinatal and NICU care. The gestational age difference with regard to ACS use, with regard to mortality (<50% and > 50%) is shown with their data but their conclusion on ACS use differ on quality of perinatal and NICU care needs more explanation and discussion as it is mentioned they provided active resuscitation to all infants born at 23-24 wks.

Authors are studying the effect of ACS with regard to mortality, it was important to have the information regarding single/multiple, or 1 or 2 doses of ACS, whether that explained some of the difference seen in this study between the groups at different GA.

Please explain what do they mean when they say “the potential bias of over-represented women admitted in advanced labor in the without ACS use group was not adjusted for in this study”

6. PLOS authors have the option to publish the peer review history of their article (what does this mean?). If published, this will include your full peer review and any attached files.

Reviewer #1: No

---

## [Author Response · Author response to Decision Letter 0]

14 Sep 2020

Response to reviewer 

Reviewer #1: This article by Park et al., on Mortality rate-dependent variations in antenatal corticosteroid-associated outcomes in very low birth weight infants with 23-34 weeks of gestation: A nationwide cohort study is an important topic with regard to use of antenatal steroids and assess why their effect is different at different GA based on mortality.

Overall, the paper is well written and is clear.

Introduction: provides sufficient background information.

Methods: The data is from 67 neonatal units from KNN registry. Rest of the methodology section is clear.

Results:

Authors have indicated ACS use was 81% in group1 for GA 23-24wk compared to group 2 was 72%.

Mortality was significantly higher for 23-24 wk. in-group 2 compared to group 1 and rest of the GA the mortality rates remained high in-group 2.

The incidence of IVH reduced in 23-24 wk., reduced PVL at 25-26 wk., decreased RDS, early onset sepsis at 29-34 wk. and increased NEC at 23-26 wk. in-group 1 with ACS use.

Symptomatic PDA at 25-26 week, RDS and sepsis at 29-34 wk. was higher in-group 2 with ACS use.

BPD/ROP rates were not different between the groups.

They have not given details of ACS with regard to doses; partial or complete between the groups on infants who received ACS

Discussion:

Authors have quoted important studies showing benefits of ACS use on mortality and morbidities in infants born at 22-34 wk. gestation.

With improved IVH, RDS, PVL, early onset sepsis in Group 1 with mortality <50% with ACS use.

Authors make a good point that with ACS use, some of the morbidities are different when compared based on mortality rate for each gestational age category.

Most morbidities were much improved in infants in group 1 and some morbidities were better (NEC) in group 2 with ACS use

They mentioned in the method section that they were unable to group them based on the neonatal units because of wide institutional variation in the mortality rate in these infants(23-24 wk). Their conclusion that the beneficial effects of ACS use differ not only GA but also by quality of perinatal and NICU care. The gestational age difference with regard to ACS use, with regard to mortality (<50% and > 50%) is shown with their data but their conclusion on ACS use differ on quality of perinatal and NICU care needs more explanation and discussion as it is mentioned they provided active resuscitation to all infants born at 23-24 wks.

Authors are studying the effect of ACS with regard to mortality, it was important to have the information regarding single/multiple, or 1 or 2 doses of ACS, whether that explained some of the difference seen in this study between the groups at different GA.

(Single/multiple or 1 or 2 doses of ACS) 

Thank you for your comment, and we fully agree with your comment that the effects of single/multiple or 1 or 2 doses of ACS on mortality should be further explained. 

Just as the reviewer has pointed out, Cochrane systemic review of 2017 (Roberts D, et al. PMID: 28321847) states that “further information is also required concerning the optimal dose-to delivery interval, and the optimal corticosteroid to use” in regards to “antenatal corticosteroids for accelerating fetal lung maturation for women at risk of preterm birth”.

It is rightly so that single/multiple or 1 or 2 doses of ACS and it effect on mortality should have been further dissected, but, the data in our study lacked such details. Therefore, we have added this limitation to the discussion and also explained it in the definition section:” ACS treatment was defined as the administration of any corticosteroid to the mother at any time before delivery to accelerate fetal lung maturity.” (Page 6, Line 117-119)(Page 14, line 18-19)

(Further explanation on difference on quality of perinatal and NICU care)

We thank the reviewer for the insightful comment. 

As the reviewer has pointed out, it is mentioned in the method section that “only VLBWIs actively resuscitated in the delivery room, and admitted to the NICU” were included in this study and those that did not received active resuscitation were excluded. Our intention was to take our included subjects and further compare to stress that ACS use itself is not the only factor affecting the outcome but, in fact, the quality of clinical practices such as well-experienced and skillful neonatologist, better early admission care, and prevention of complication can be of greater influence in increasing survival in 23-24 wk. 

We have added “Taken together, these results suggest that improved quality of perinatal and NICU care such as well-experienced and skillful neonatologist, better early admission care, and prevention of complication improved mortality of peri-viable infants at 23-24 weeks gestation. Therefore, higher quality of clinical care can be a prerequisite to the beneficial effects associated with ACS use.” in the discussion section of revised manuscript. (Page 14, line 10-14)

Please explain what do they mean when they say “the potential bias of over-represented women admitted in advanced labor in the without ACS use group was not adjusted for in this study”

(Bias)

We thank reviewer’s thoughtful comments. 

We thank the reviewer’s thoughtful comments. The statement “ the potential bias of over-represented women admitted in advanced labor in the without ACS use group was not adjusted for in this study” was included to point out the bias of hospitals that might have had worse outcome due to greater proportion of mothers in immediate labor who did not have enough time to administer ACS. As the reviewer pointed out, this statement is highly prone to confusion and is not absolutely crucial to point out; therefore, we have removed the statement in revised manuscript.(Page 14, line 19) 

Thank you.

We appreciate the thoroughness with which the reviewer examined our paper, and hope that it is now suitable for publication in the prestigious journal “PLOS ONE.” 

Sincerely, 

Won Soon Park, MD. PhD.

---

## [Editor Report · Decision Letter 1]

22 Sep 2020

Mortality rate-dependent variations in antenatal corticosteroid-associated outcomes in very low birth weight infants with 23-34 weeks of gestation: A nationwide cohort study

PONE-D-20-16654R1

Dear Dr. Won Soon Park,

We’re pleased to inform you that your manuscript has been judged scientifically suitable for publication and will be formally accepted for publication once it meets all outstanding technical requirements.

Kind regards,

Georg M. Schmölzer

Academic Editor

PLOS ONE
---

## [Editor Report · Acceptance letter]

25 Sep 2020

PONE-D-20-16654R1 

Mortality rate-dependent variations in antenatal corticosteroid-associated outcomes in very low birth weight infants with 23-34 weeks of gestation: A nationwide cohort study 

Dear Dr. Park:

I'm pleased to inform you that your manuscript has been deemed suitable for publication in PLOS ONE. Congratulations! Your manuscript is now with our production department. 

Kind regards, 

on behalf of

Dr. Georg M. Schmölzer 

Academic Editor

PLOS ONE